# Differences in arousal and valence on the Korean phoneme of artificial voice between Korean and Chinese women

**Kwang Jin Lee**[1☺], **Gi-Eun Lee**[2☺], **San Ho Lee**[1☺], **Jang-Han Lee**[2☺]*

1 Department of European Language and Cultures, Chung-Ang University, Seoul, Republic of Korea,
2 Department of Psychology, Chung-Ang University, Seoul, Republic of Korea

☺ These authors contributed equally to this work.
* clipsy@cau.ac.kr

## Abstract

### Background

Although active research is in progress in the fields of psychology and linguistics on the emotional characteristics of the symbol and meaning of sound itself, since the systematic emotional model is not applied, each researcher uses a subjective concept and acts as an obstacle to the expansion of research. There is a limitation in that it cannot be confirmed whether the sound symbol has universality regardless of cultural differences between different languages.

### Methods

In this study, the difference between the arousal and valence of emotions felt toward Hangul phonemes was explored according to consonant and vowel through comparison between Korean and Chinese women. 38 Korean women and 32 Chinese women were recruited, and an online experiment was conducted in which arousal and valence were reported for 42 Hangeul phoneme sound stimuli.

### Results

As a result of comparing the arousal and valence of each group, Koreans showed significantly higher arousal scores than Chinese, and these results showed different differences according to consonant and vowel. In valence, there was a difference between nationalities only according to consonant indicating that Koreans showed lower positivity toward aspirated sounds than Chinese. Through these results, it was confirmed that the emotional meaning of the sound symbol between different languages is different, which can be affected by consonant and vowels.

### Conclusion

This study identified differences in emotional perception between cultures by using two dimensions of emotions, arousal, and valence, which are systematized for sound symbols,

**Data Availability Statement:** All relevant data are within the paper and its Supporting information files.

**Funding:** This work was supported by the Ministry of Education of the Republic of Korea and the

National Research Foundation of Korea (NRF-2022S1A5C2A04093121).

and suggests implications for the relationship between sound symbol and emotions and cultural differences in the future.

## Introduction

Rapid identification and delivery of emotional elements in sound stimuli are very important factors for survival and adaptation in human verbal communication [1]. Emotional sound symbolism describes this as an adaptation account chosen for its adaptive value [2]. Since it is an important point directly related to survival to receive emotional signals quickly and effectively to human beings who communicate through language, if we find relatively similar and common emotions in phonemes, which are the smallest units that people can feel, we can facilitate important emotional responses that are required in the environment more easily. Therefore, it remains a strong link between phonemes and emotional meaning in our conversations across cultures today and remains a topic of linguistic and psychological research in communication.

From a psychological point of view, sound symbolism has been conducting research on the sensory information recognized and perceived by people through specific sounds and their emotional meaning. Researchers have attempted to find emotional meanings by linking hearing with perception areas for various senses, such as sight [3, 4], taste [5], smell [6], touch [7], and motion [8]. One of the most famous examples in the visual field, the "bouba/kiki effect" is a phenomenon in which people associate a round shape when they hear "bouba" and a pointed shape when they hear "kiki" [1]. This effect is observed regardless of the mother tongue and age. Another example is the study of flavors that people expect chocolate with unvoiced sounds name to be sweeter [5], and the smell of choosing fake words that contain unpleasant smells [6]. These findings suggest that people have a relatively common specific feeling through synesthetic perception by simply using a sound stimulus.

These sounds stimuli correspond to nonverbal elements of communication, suggesting that they are closely related to emotion, and Feldman, Philippot & Custrini [9] suggests that nonverbals are more effective indicators of understanding the emotions of the opponent than language. However, the question remains whether one specific sound can give people the same meaning and emotion regardless of culture. Emotional sound symbolism assumes that the emotional universality of these sounds will be shared across cultures [2, 10]. For example, in thousands of languages, the word 'dog' tends to use consonants and vowels with sounds that can represent relatively large entities, while consonants and vowels with smaller sounds can be used to symbolize relatively small entities, such as 'ant'. This claim has been supported empirically. However, as it is argued that differences in popular culture or generational experiences give a significant difference to sound symbolism [11], there is evidence that more specific attributes can be applied to a specific population group than to other population groups. In addition, there are clues that cultural differences between language and emotional meaning can have an effect through research that there is a difference in preference for emotional words depending on cultures [12]. Cultural differences can affect the perception of morphemes with meaning (eg, happiness, anger, fear), but the smallest sound unit that does not contain meaning (eg, /ka/, /nu/, /si/). It is easy to expect that cultural differences will not have an impact on however, considering the basic assumption of sound symbolism that sounds can affect the perceived emotional meaning of a person due to the characteristics of sound itself, practical verification is required whether cultural differences do not affect such emotional perception.

As the most basic classification system for emotions that people feel through perceived information, there is a valence and arousal model of emotions. This is well-known as Russell's Circumplex model, and it is a model that expresses emotion using two axes of emotion dimension: arousal and valence [13]. Valence represents the continuity of positive and negative stimuli, pleasure and displeasure, and arousal refers to the perceived intensity of a stimulus from very excited to very calm [14, 15]. Russell believed that emotions such as happiness, sadness, and fear had a specific distribution in the two dimensions of emotion. Since these distributions vary slightly with individual and cultural differences, he proposed only approximate distributions of emotions [13].

Introduction of the most basic system for emotion research in sound symbolism can bring about the generalization of sound symbolism research results and the expansion of the research field. According to the existing claim that all sounds have acoustic characteristics such as 'rough or soft', 'strong or weak', various measurements have been made on how a sound is felt, including whether it feels rough or soft. However, the adjective meanings of sounds used in the study, such as 'strong or weak', 'attractive or ugly', were not integrated. While this classification can certainly be attractive and intuitive, it makes it difficult to accurately classify the emotional characteristics people feel through sound. The current unstructured research on sound symbolism hinders the reliability of research results and the repeated validation of the research due to the arbitrary naming and setting of individual researchers on emotions. This is because, as Russell argues, each study can lead to different results due to individual and cultural differences involved in emotional words. This will help clarify the somewhat ambiguous emotional expressions (e.g., cheerful, dark, soft) currently used in sound symbolism and further reveal the influence of individual differences and cultural differences on sound symbolism. Therefore, we tried to make an exploratory attempt to expand the research area through the systematization of the emotional dimension of sound symbolism.

Artificial intelligence (AI), a key technology of the 4th industrial revolution, is being used for various purposes. On top of that, it has become possible to communicate between humans and AI through the voice user interface (VUI), one of the important functions of AI. However, the challenge remains whether AI speakers can become a platform that can interact more perfectly with humans beyond providing simple convenience functions such as weather notifications and alarm settings. For more complete communication, research on "emotional awareness" and "emotional expression" of AI is being actively conducted. The purpose of most studies was whether AI could recognize the user's emotions and express emotions like humans according to the user's emotional state. However, for communication, not only the emotion of AI in both directions, but also the human perception of what is transmitted from AI speakers is important.

Today, research on sound symbolism has been conducted on the emotional elements of sounds by dividing the characteristics of sounds into various categories (e.g., vowel, consonant and tone). Most of the research on sound symbolism done so far has focused on vowels rather than consonants [16, 17]. Due to the characteristics of consonants in that they cannot be separated from vowels and there are many differences between languages compared to vowels, very few studies have been conducted using consonants [18]. However, when considering the practical use of language, ignoring the influence of consonants, and focusing only on the transformation of vowels may act as a limit in perceiving emotional meaning. Research results suggest that the impression perceived by the listener is different depending on the consonants, which supports this claim [8]. Therefore, in this study, we tried to measure the emotional value of consonants through the classification according to the consonant by fully utilizing the consonants that have been neglected in sound symbolism. The classification of consonants varies depending on the language, but a method of classifying plain consonants, aspirated

consonants, and voiced consonants, which is a common consonant in many languages including English and Korean, and known to give a similar emotional impression, was adopted [19].

The purpose of this study is to explore arousal and valence, which are basic units of emotion that people feel about Hangul phonemes, through comparison with other cultures. Accordingly, an experimental study was conducted on Korean and Chinese women living in Korea who could speak Korean using the Hangul phoneme, the most basic unit, as an experimental stimulus. This study was conducted with an exploratory hypothesis that there would be significant differences according to consonant and vowels according to cultures, rather than testing previously established hypotheses, depending on the purpose of the study to see the differences in characteristics.

## Methods

### Participants

A total 86 of Participants was recruited from an online bulletin board on university websites. All participants were told that they would be recruited for psychological experiments on phonemes and emotions. The entire process of the experiment was conducted online. A power analysis using G*Power 3.1.9.7 (University of Dusseldorf, Dusseldorf, Germany) showed that using an effect size 0.25, alpha error probability of 0.05, power 0.95, number of groups 2, number of measurements 6, correlation estimator value among pairs of the repeated measurements 0.5 and non-sphericity correction 1, minimum sample size was 28 (each condition 14). To clearly see the difference in variables, the number of participants was limited to women, considering that women showed greater emotional responsiveness than men. Inclusion criteria of participation were: 1) more 20 years of age female, 2) being Korean or Chinese nationality (Mandarin Chinese), 3) must be able to speak Korean. All participants were provided with informed consent and were informed that they can terminate the experiment at any time prior to participation in the present study and that they would be paid $ 20 for the participation.

### Materials

**Stimuli.** In this experiment, 42 Hangul phoneme stimuli were made and used with artificial human sounds using the TTS (Text To Speech) program. All stimuli were made equally in length of 500ms. Considering the most basic phonological composition and fatigue of participants due to a long experiment time, Hangul's double consonants were excluded, and only three vowels were used. Three vowels with the greatest differences in phonetic structure were selected [20]. In addition, 14 consonants were classified into three types according to the classification system of consonant. It was classified as lenis with low tension, aspirated consonant produced by bursting strong air when plosives were produced, and voiced consonant accompanied by the resonant of the vocal cords. A total of 42 voice stimuli were used as a combination of 14 Korean consonants and 3 Korean vowels. (see Table 1).

**Questionnaires.** *Positive and Negative Affect Schedule scale (PANAS)*. Positive and Negative Affect Schedule scale (PANAS) was required to assess positive affect and negative affect [21]. It consists of 20 items, of which ten items reflect expectations for positive affect (PANAS-P) and ten for negative affect (PANAS-N). Participants were asked to evaluate to what extent they currently feel that way on a 5-point Likert scale ('very slightly or not at all' to 'extremely'). Scores for each subscale can range from 1 to 5, with higher scores representing higher levels of positive affect and negative affect. In this study, Korean version of PANAS (K-PANAS [22]) and Chinese version of PANAS (C-PANAS [23]) were used. Cronbach's alpha of K-PANAS-P was .694 and K- PANAS-N was .761, and Cronbach's alpha of C-PANAS-P was .866 and C- PANAS-N was .877.

**Table 1. Manners of consonant of English and Korean alphabet.**

| Consonant | Korean Code | English Expression | English phonics | Combination of consonants and vowels | | |
|---|---|---|---|---|---|---|
| | | | | **Vowel** | | |
| | | | | **[a]** | **[u]** | **[i]** |
| lenis | ㄱ | Giyeok | [k, g, k^] | 가 | 구 | 기 |
| | ㄷ | Digeut | [t, d, t^] | 다 | 두 | 디 |
| | ㅂ | Bieup | [p, b, p^] | 바 | 부 | 비 |
| | ㅅ | Siot | [s, ʃ] | 사 | 사 | 시 |
| | ㅈ | Jieut | [tʃ, ʤ, t^] | 자 | 주 | 지 |
| | ㅎ | Hieut | [h] | 하 | 후 | 히 |
| aspirated | ㅊ | Chieut | [tʃh, t^] | 차 | 추 | 치 |
| | ㅋ | Kieuk | [kh, k^] | 카 | 쿠 | 치 |
| | ㅌ | Tieut | [th, t^] | 타 | 투 | 티 |
| | ㅍ | Pieup | [ph, p^] | 파 | 푸 | 피 |
| Voiced | ㄴ | Nieun | [n] | 나 | 누 | 니 |
| | ㄹ | Rieul | [r, l] | 라 | 루 | 리 |
| | ㅁ | Mieum | [m] | 마 | 무 | 미 |
| | ㅇ | Ieung | [ŋ] | 아 | 우 | 이 |

*State-Trait Anxiety Inventory (STAI)*. It is a 40-item self-report measure of state and trait anxiety [24]. The higher scores represent more intense or more frequent feelings of anxiety. The STAI is composed of 20 self-report items rated on a 4-point Likert scale (1 = not at all to 4 = very much so); both state and trait versions have scores ranging from 20 to 80. The trait version (STAI-T) measures 'trait anxiety', a general, long-term form of anxiety, while the state version (STAI-S) measures "state anxiety", or a temporary form of anxiety. In this study, Korean versions of STAI-T and STAI-S (K-STAI [25]) and Chinese version of STAI-T and STAI-S (C-STAI [26]) were used. Cronbach's alpha of K-STAI-T was .931 and K-STAI-S was .949, and Cronbach's alpha of C-STAI-T was .800 and C-STAI-S was .757.

*Center for Epidemiologic Studies Depression Scale (CES-D)*. Center for epidemiologic studies depression scale (CES-D) was required to examine participant's baseline level of depression mood in case of when emotions can feel more intense or flat [27]. Participants were asked how often they felt a certain way over the past week, with four response options ranging from 0 (rarely) to 3 (all the time). In this study, Korean version of CES-D (K-CES-D [28]) and Chinese version of CES-D (C-CES-D [29, 30]) were used. Cronbach's alpha of K-CES-D was .936 and C-CES-D was .921.

## Procedure

This study was approved by the University Institutional Review Board (NO.1041078-202211-HR-256). This study used an online questionnaire site, "SurveyMonkey", considering accessibility to stimuli and convenience of participants. The questionnaire was provided in the order of 'Agreement-Demographic Variables-Affective ratings of phoneme-Psychological Variables-Debriefing', and it took about one hour. For the convenience of participants, we made it possible to respond using mobile phones, tablets, and computers. All participants were able to start an online survey after agreeing to participate in the survey. In the first page of online survey form, they were informed about the research procedure, the possible risks, and the right to voluntarily stop participating in the experiment in writing. Only participants who read the

written consent form and agreed to participate were allowed to access the next stage of the questionnaire and survey. First, set one hour to perform the experiment (not to stop this experiment in the middle). Second, set the surrounding environment so that it is not affected by external stimuli (e.g., noise). Next, recommend using earphones and check the volume of computer of laptop in advance. Lastly, follow the on-screen instructions. The 42 auditory stimuli were presented randomly, and they rated the degree of arousal and valence after hearing each stimulus. They could hear as much as they wanted by clicking the button on the screen. Participants who completed the online survey were given an e-coupon worth $20 through the collected contact information.

## Data analysis

A total of 70 data were analyzed in the final analysis, excluding 16 participants, a condition corresponding to the exclusion criteria among 86 participants (e.g., male, less 20yr, other nationality). 2 (Nationality; Korea, China) × 3 (Consonant; lenis, aspirated, voiced) × 3 (Vowel; /a/, /u/, /i/) mixed Analysis of Variance (ANOVA) was conducted to explore the pattern of arousal and valence. Bonferroni post-hoc test was performed to identify significant difference among nationality, consonant, and vowel. Independent samples T-test was performed for continuous variables to compare demographic and psychological characteristics. Statistical analysis was performed with SPSS 26.0 for windows (SPSS inc., Chicago, USA).

## Results

### Sample characteristics

The final sample of this study consisted of 70 participants ranging in age from 18~43 years old ($M$ = 23.81, $SD$ = 3.77). There were no significant differences between groups in age [$t$ (68) = -1.097, $p$ = .279], scores of PANAS-P [$t$ (68) = -1.217, $p$ = .228], PANAS-N [$t(68)$ = -674, $p$ = .099], STAI-T [$t$(68) = -1.357, $p$ = 179], STAI-S [$t$(68) = -.921, $p$ = .360], and CES-D [$t$(68) = -.610, $p$ = .544] (see Table 2).

### Analysis of the difference in arousal according to nationalities, consonants, and vowels

To examine that there are differences between nationalities for consonants and vowels in arousal, 2(Nationality; Korea, China) × 3 (Consonant; lenis, aspirated, voiced) × 3 (Vowel; /a/, /u/, /i/) three-way mixed ANOVA was conducted. The dependent measure was the amount of

**Table 2. Mean (SD) of sample characteristics.**

| Measure | Korean (N = 38) | Chinese (N = 32) | Test Statistics ($d / \chi^2$) |
|---|---|---|---|
| Age (yrs) | 25.24(2.84) | 26.22(4.61) | -1.091 |
| PANAS-P | 27.34(4.35) | 28.91(6.56) | -1.217 |
| PANAS-N | 24.16(5.60) | 26.75(7.34) | -1.674 |
| STAI-T | 45.08(7.25) | 47.25(5.90) | -1.357 |
| STAI-S | 43.55(7.79) | 45.22(7.23) | -.921 |
| CES-D | 16.82(6.78) | 18.00(9.43) | -.610 |

*Note*. Mean (standard deviation); PANAS-P: Positive and Negative Affect Schedule Scale-Positive; PANAS-N: Positive and Negative Affect Schedule Scale-Negative; STAI-T: State-Trait Anxiety Inventory-Trait anxiety; STAI-S: State-Trait Anxiety Inventory-State anxiety; CES-D: Center for Epidemiologic Studies Depression Scale.

**Table 3. Mean (SD) of arousal and valence of consonant and vowel between group.**

| Dependent measure | Consonant | vowel | Korean (N = 38) | Chinese (N = 32) | Test Statistics (F) | Partial eta squared ($\eta^2$) |
|---|---|---|---|---|---|---|
| Arousal | lenis | /a/ | 4.86(1.30) | 4.12(1.64) | 4.309* | .060 |
| | | /u/ | 4.64(1.23) | 4.20(1.71) | 1.503 | .022 |
| | | /i/ | 5.45(1.24) | 4.59(1.84) | 5.368* | .073 |
| | aspirated | /a/ | 5.89(1.46) | 4.63(2.07) | 8.777** | .114 |
| | | /u/ | 5.59(1.42) | 4.76(1.88) | 4.460* | .062 |
| | | /i/ | 5.53(1.54) | 4.26(1.99) | 9.115** | .118 |
| | voiced | /a/ | 4.56(1378) | 3.77(1.79) | 3.947 | .055 |
| | | /u/ | 4.01(1.36) | 3.76(1.89) | .431 | .006 |
| | | /i/ | 4.59(1.20) | 4.00(1.69) | 2.921 | .041 |
| Valence | lenis | /a/ | 5.63(.83) | 5.99(1.48) | 1.072 | .024 |
| | | /u/ | 5.21(1.16) | 5.58(1.60) | 1.178 | .017 |
| | | /i/ | 5.19(1.28) | 5.50(1.52) | .874 | .013 |
| | aspirated | /a/ | 4.91(.91) | 5.88(1.61) | 10.110** | .129 |
| | | /u/ | 4.74(1.09) | 5.57(1.47) | 7.399** | .098 |
| | | /i/ | 4.70(1.18) | 5.89(1.46) | 14.160*** | .172 |
| | voiced | /a/ | 6.34(1.48) | 6.19(2.05) | .126 | .002 |
| | | /u/ | 5.80(1.37) | 5.48(1.64) | .754 | .011 |
| | | /i/ | 5.75(1.19) | 5.77(1.74) | .002 | .000 |

*Note*. Mean (standard deviation).

*$p < .05$.

**$p < .01$.

***$p < .001$.

arousal rated by how much a participant was aroused after hearing voice stimuli. There were significant interaction effects on the nationality and consonant [$F(2, 136) = 3.648$, $p = .029$, $\eta^2 = .051$], nationality and vowel [$F(2, 136) = 3.199$, $p = .044$, $\eta^2 = .045$], and consonant and vowel [$F(4, 272) = 7.291$, $p = .000$, $\eta^2 = .097$] (*see* Table 3). In Bonferroni post-hoc test on nationality and consonant, Korean reported higher arousal than Chinese in lenis [$F(1, 68) = 4.477$, $p = .038$, $\eta^2 = .062$] and aspirated consonant [$F(1, 68) = 8.755$, $p = .004$, $\eta^2 = .114$], whereas there was no distinct difference between nationalities in voiced consonant [$F(1, 68) = 2.663$, $p = .107$, $\eta^2 = .028$]. Korean reported higher arousal than Chinese in /a/ [$F(1, 68) = 7.458$, $p = .008$, $\eta^2 = .099$] and /i/ [$F(1, 68) = 2.320$, $p = .132$, $\eta^2 = .033$], whereas there was no distinct difference between nationalities in /u/ [$F(1, 68) = 7.409$, $p = .008$, $\eta^2 = .098$] in Bonferroni post-hoc test on nationality and vowel (Fig 1). And as shown in Fig 2, Korean reported higher arousal than Chinese in lenis [$F(1, 68) = 4.477$, $p = .038$, $\eta^2 = .062$] and aspirated consonant [$F(1, 68) = 8.755$, $p = .004$, $\eta^2 = .114$], whereas there was no distinct difference between nationalities in voiced consonant [$F(1, 68) = 2.663$, $p = .107$, $\eta^2 = .028$]. There were no significant three-way interaction effects between nationality, consonant, and vowel [$F(4, 272) = .147$, $p = .964$, $\eta^2 = .002$].

## Analysis of the difference in valence according to nationalities, consonants, and vowels

To examine that there are differences between nationalities for consonant and vowels in valence, 2 (Nationality; Korea, China) × 3 (Consonant; lenis, aspirated, voiced) × 3 (Vowel; /a/, /u/, /i/) mixed ANOVA on valence was conducted. The dependent measure was the amount

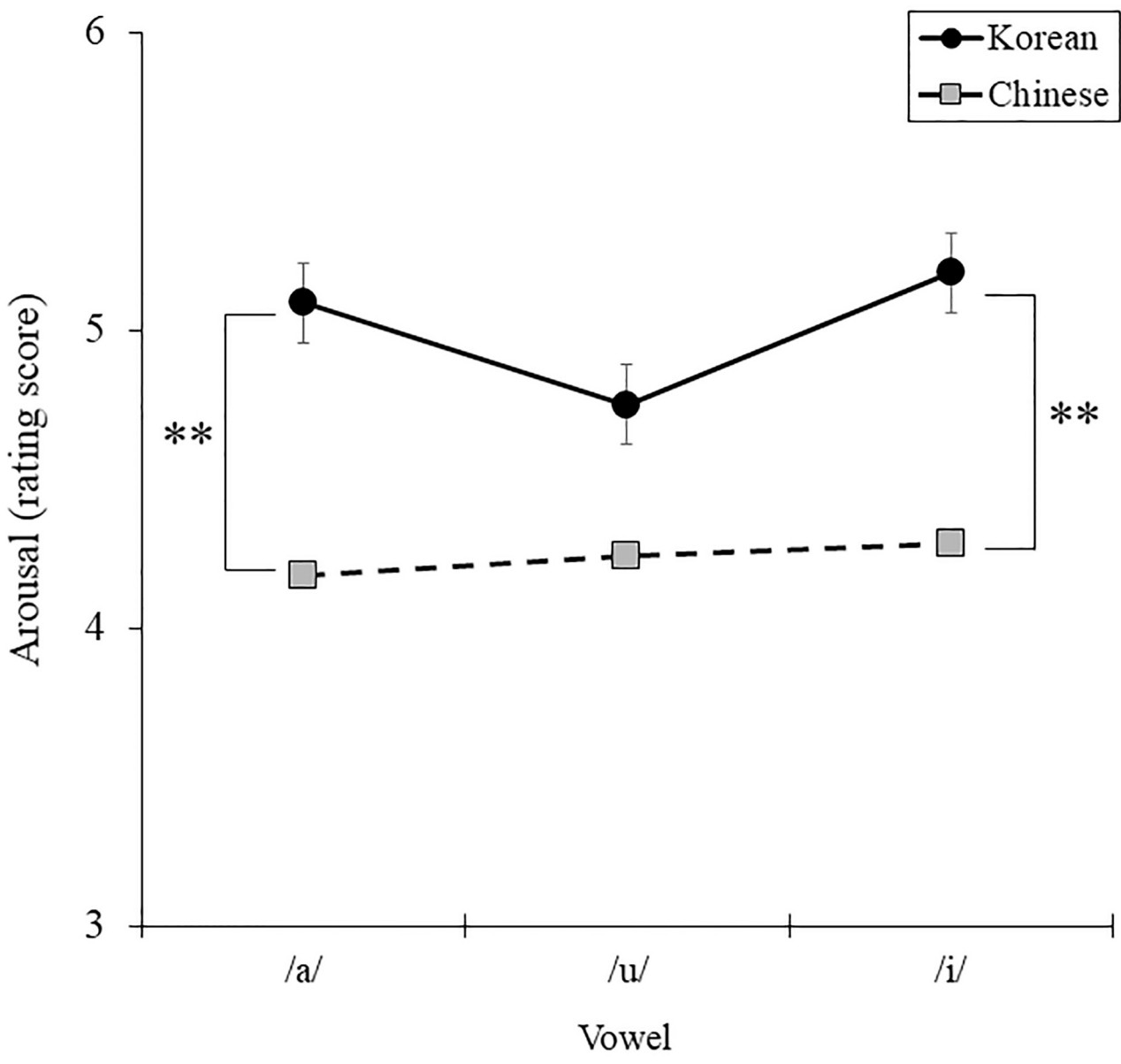

**Fig 1. Comparison of arousal for nationality groups in vowel.**

of valence rated by how much a participant felt positively after hearing voice stimuli. There was a significant interaction effect on the nationality and consonant [$F_{(2, 136)}$ 18.355, $p = .000$, $\eta^2 = .213$], but no significant in nationality and vowel [$F_{(2, 136)} = .599$, $p = .551$, $\eta^2 = .009$], and consonant and vowel [$F_{(4, 272)} = 1.725$ $p = .145$, $\eta^2 = .025$] (*see* Table 3). In Bonferroni post-hoc test on nationality and consonant, Chinese reported higher positive than Korean in aspirated consonant [$F_{(1, 68)} = 14.809$, $p = .000$, $\eta^2 = .179$], whereas there was no distinct difference between nationalities in lenis [$F_{(1, 68)} = 1.570$, $p = .215$, $\eta^2 = .023$] and voiced consonant [$F_{(1, 68)} = .204$, $p = .653$, $\eta^2 = .003$]. There were no significant three-way interaction effects between nationality, consonant, and vowel [$F_{(4, 272)} = .344$, $p = .848$, $\eta^2 = .005$] (Fig 3).

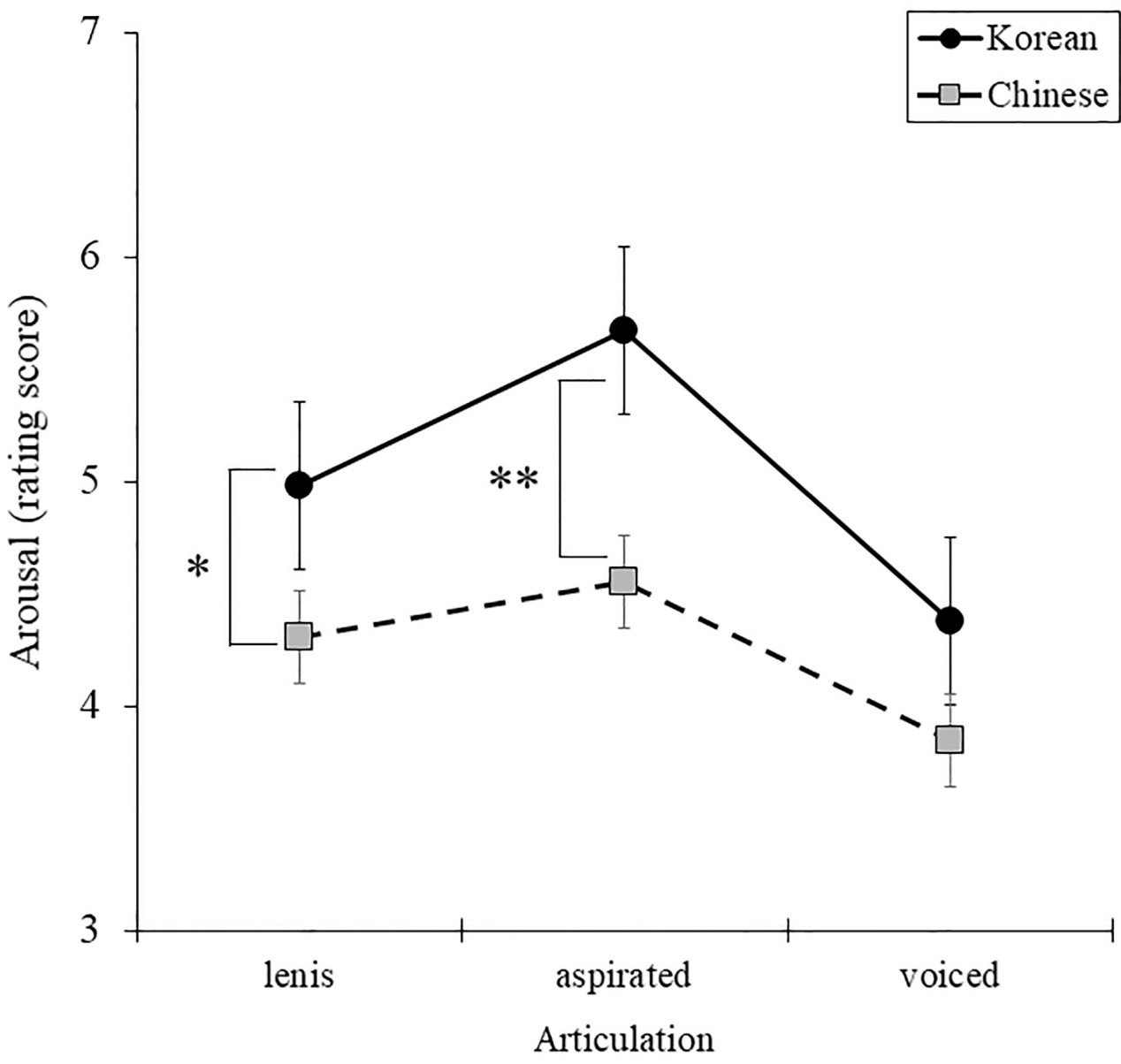

**Fig 2. Comparison of arousal for nationality groups in consonant.**

## Discussion

This study explored cultural differences in emotional perception by categorizing the two-dimensional arousal and emotional valence of Korean and Chinese women according to consonants and vowels. The results of this study showed that there are cultural-specific differences in the arousal and emotional valence according to the consonant and vowel in Korean and Chinese. This difference was particularly pronounced in the arousal song, and the valence song showed different aspects between nationalities according to the consonant.

The most important finding of this study, conducted with an exploratory purpose, was that it revealed that there was a difference in the emotional perception of phonemes between Korean and Chinese women, especially in the degree of arousal. This result is consistent with

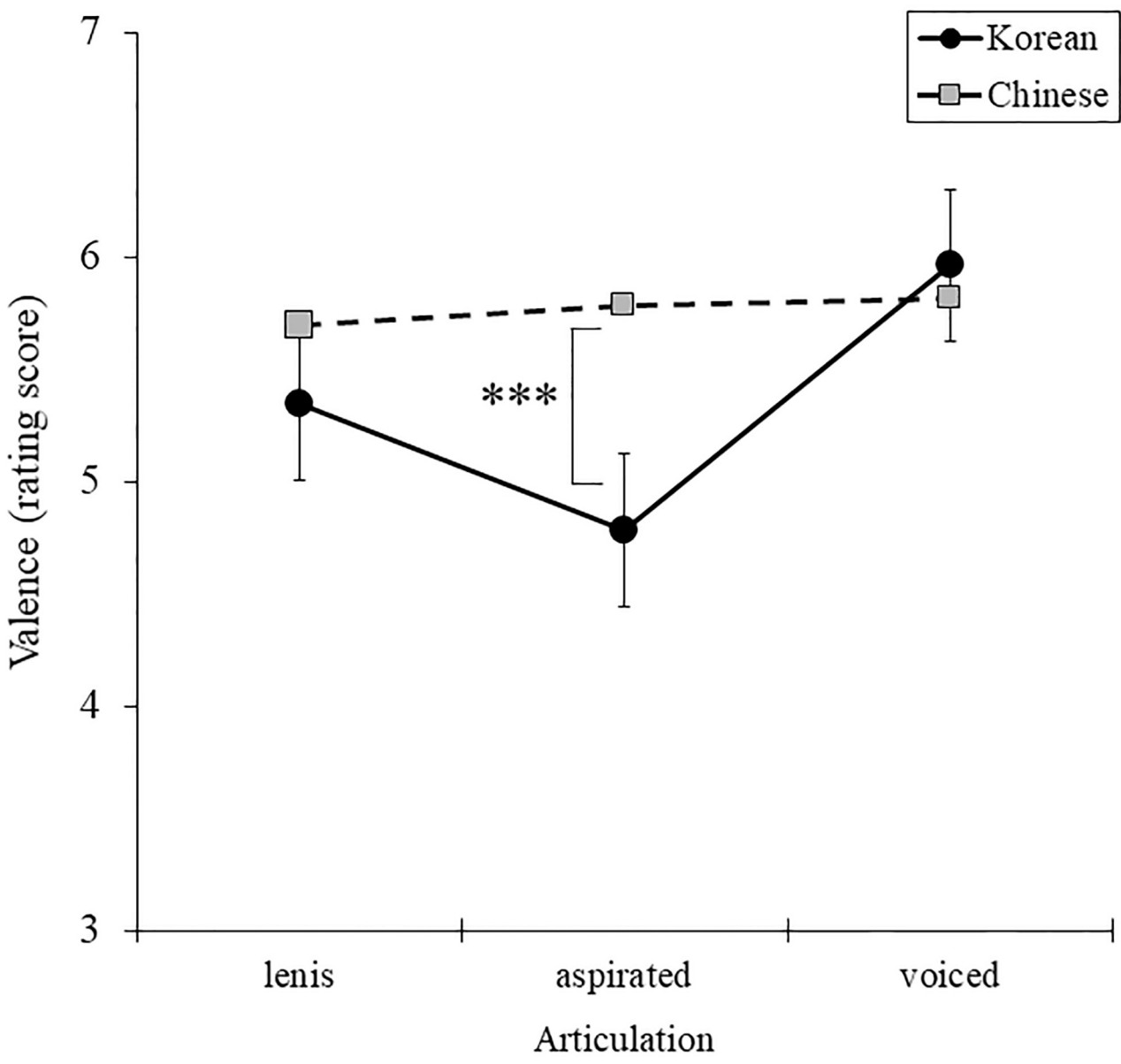

**Fig 3. Comparison of valence for nationality groups in consonant.**

Russell's argument that there may be individual and cultural differences in emotional arousal [13]. Through this study, it was experimentally found that this principle can be applied to sound symbolism, and different levels of arousal can be reported by hearing the same phoneme. This difference is meaningful in that it not only revealed that there is a difference between nationalities, but also that it can occur depending on the consonant and the category of vowels. This result is contrary to the claim of cultural universality of sound symbolism.

What we should note is the result of the interaction between nationality and consonant in both arousal and emotion. This is interpreted because of reflecting routes different from the acoustic symbolism that have been conducted mainly on vowels. According to the results of this study, it is true that vowels make a significant difference in emotional meaning in sound

symbolism as previously claimed, but the main factor that can cause differences according to different languages is the consonant. Existing criticism that consonants cannot be separated from vowels is a valid and important point, but the results of this study, which at least coincided with the categories of vowels, showed that the difference in perception of emotional meaning according to language was due to the consonant. In fact, if vowels are actively studied in perceptual domains such as size perception, it is inferred that the consonant can have a more direct relationship with emotion because it is known to form a sense of sound.

We were forced to note the difference in results from each arousal and valence. The point is that the arousal showed a more clear and distinct difference, whereas in the valence, only the interaction between nationality and consonant was significant. As an explanation for this point that except for the presence of the voice, which is a super segment element, the biggest difference between Korean and Chinese syllables is in phoneme combination constraints. The syllable consonants that can make the end sound in Korean are 7 of ㅂ/p/, ㄷ/t/, ㄱ/k/, ㅁ/m/, ㄴ/n/, ㅇ/ ŋ/, ㄹ/l/, while Chinese is only 2 of /-n/ and / ŋ/. The difference in the phoneme combination constraints of syllables is the cause of the biggest difference between Korean and Chinese phonological variations. In addition, in the results of this study, Koreans showed higher arousal than Chinese in lenis and aspirated consonants, while there was no clear difference between nationalities in voiced consonants. In general, Mandarin Chinese does not use voiced and voiceless sounds as discriminator as in Korean, so how well Chinese and Korean distinguish between voiced and aspirated sounds does not seem to affect the valence of audible sounds. However, the response values to the voiced consonants collected in this study tended to be somewhat lower in both Koreans and Chinese compared to lenis and aspirated consonants. In other words, it is estimated that the sound accompanying the vocal cords like voiced consonants lowered the arousal level, while it played a role in activating valence level.

This study has some limitations. The first is that comparisons with other languages have not been made. It should be noted that even sounds with the same consonant may have subtle differences in various languages such as English, French, Chinese, and Japanese. In particular, in the case of Chinese, there are various dialects in addition to national and cultural differences, so research to subdivide each language characteristic is needed in the future. This includes not only the language, but also the elements of the population of various cultures other than Korea and China. In future research, we propose a comparison with these various languages and cultures. Second, the subdivision of consonant and vowel classification was not achieved. We used three classifications of plain consonants, aspirated consonants, and voiced consonants with reference to existing literature, but consonant can be further subdivided. Therefore, in future study, it is necessary to compare and analyze the difference between the valence and arousal values of language in consideration of a more detailed articulation classification. Because of the clarity of emotional value and the fact that the number of vowels varies greatly in each language, only three vowels were used in this study, /a/, /u/, and /i/, but in real life, more diverse vowels are used do. Since other interesting results can be obtained depending on such detailed classification, it is suggested that the difference according to various consonant can be studied in future studies. Third, this study was conducted on women only. According to the existing claims that women are more emotionally reactive and that gender differences in sound symbolism were not significant, this study only conducted a study on women, but there may be interactions with gender due to cultural differences. Future research suggests that the addition of gender to cultural differences may lead to other interesting results. In addition, in this study, the difference in the length of time living in Korea was not sufficiently controlled. Even if participants started living in Korea after becoming an adult, there may be differences in response depending on the period of contact with Korean culture, so it will be necessary to consider in future studies. Finally, the voice used in this study was a female

voice and was artificially produced. It should be noted that different results may be obtained if the gender of the voice is different or if a real human voice instead of an artificial voice is used, and a repeat verification procedure for these differences will be required in a follow-up study.

Despite these limitations, this study has the implications of systematizing emotions according to two dimensions in sound symbolism and confirming that cultural differences can affect perceived emotions. This suggests a more systematized method for the emotions used in the sound symbolism research area, and at the same time suggests both the properties of universality and cultural specificity in sound symbolism. Through the results of this study, it is suggested that a step in understanding human communication may have been achieved, further extending the sound symbolism for more complex emotions (e.g., disgust, happy, fear) beyond arousal and valence.

## Supporting information

**S1 Data.**
(XLSX)

## Acknowledgments

The authors would like to thank all participant for their contribution to this paper.

## Author Contributions

**Conceptualization:** Kwang Jin Lee, San Ho Lee, Jang-Han Lee.

**Data curation:** Kwang Jin Lee, Jang-Han Lee.

**Formal analysis:** Kwang Jin Lee, Gi-Eun Lee, Jang-Han Lee.

**Funding acquisition:** San Ho Lee, Jang-Han Lee.

**Investigation:** Kwang Jin Lee, Jang-Han Lee.

**Project administration:** San Ho Lee, Jang-Han Lee.

**Resources:** Gi-Eun Lee.

**Supervision:** Kwang Jin Lee, San Ho Lee, Jang-Han Lee.

**Writing – original draft:** Kwang Jin Lee.

**Writing – review & editing:** Kwang Jin Lee, Gi-Eun Lee, Jang-Han Lee.

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
