## [Decision Letter · Decision Letter 0]

31 Aug 2022

PONE-D-22-15241Differences in Arousal and Valence on the Korean Phoneme of Artificial Voice between Korean and Chinese WomenPLOS ONE

Dear Dr. Lee,

Thank you for submitting your manuscript to PLOS ONE. After careful consideration, we feel that it has merit but does not fully meet PLOS ONE’s publication criteria as it currently stands. Therefore, we invite you to submit a revised version of the manuscript that addresses the points raised during the review process.

We look forward to receiving your revised manuscript.

Kind regards,

Tai Ming Wut

Academic Editor

PLOS ONE

Journal Requirements:

 “This work was supported by the Ministry of Education of the Republic of Korea and the National Research Foundation of Korea(NRF-2019S1A5C2A04082405)”

“This work was supported by the Ministry of Education of the Republic of Korea and the National Research Foundation of Korea(NRF-2019S1A5C2A04082405)”

“This research was supported by the Ministry of Education of the Republic of Korea and the National Research Foundation of Korea (NRF-2019S1A5C2A04082405)”

7. We note that you have referenced (ie. Bewick et al. [5]) which has currently not yet been accepted for publication. Please remove this from your References and amend this to state in the body of your manuscript: (ie “Bewick et al. [Unpublished]”) as detailed online in our guide for authors

Reviewers' comments:

Reviewer's Responses to Questions

**Comments to the Author**

1. Is the manuscript technically sound, and do the data support the conclusions?

Reviewer #1: Yes

Reviewer #2: Partly

2. Has the statistical analysis been performed appropriately and rigorously? 

Reviewer #1: I Don't Know

Reviewer #2: Yes

3. Have the authors made all data underlying the findings in their manuscript fully available?

Reviewer #1: Yes

Reviewer #2: No

4. Is the manuscript presented in an intelligible fashion and written in standard English?

Reviewer #1: Yes

Reviewer #2: Yes

5. Review Comments to the Author

Reviewer #1: Overview and general recommendation:

In this study, the difference between the arousal and valence of emotions felt toward Hangul phonemes was explored according to consonant and vowel through comparison between Korean and Chinese women. This is an interesting study and the authors have collected a unique dataset using rigorous methodology. The design of two groups of participants from different language backgrounds makes the experiment reasonable for the purpose. I believe the authors conducted the experiment carefully in line with the strict procedure. Overall the paper is generally well written and structured. Thus, the current study is a topic of relevance and general interest to the reader of the journal.

Nonetheless, I found some of the description of the paper to be irrelevant while the descriptions of some points were inaccurate. Given this, I recommend that a minor revision is warranted. I explain my concern and raise my questions in more detail below.

Minor comments:

1. In the abstract and other part of your paper, I suggest to use consonant, instead of articulation or articulation methods to avoid confusion. In the field of Phonetics Phonology, the term ‘articulation’ refers to the configuration which produces the sounds of speech while both consonant and vowel belong to phoneme category.

2. Page 2, line 4 of the countdown. “contain meaning (eg, /ka/, /nu/, /si/)”, punctuation missing?

3. Page 2, line 3 of the countdown. “impact on However,”, typo?

4. Page 4, line 8-9 of the countdown. From a linguistic point of view, it is more appropriate to categorized them with vowel, consonant and tone). Articulation methods seem strange. You are suggested not to use this term as ‘consonant’ is more accurate, in accordance with vowel and other segmental or supra-segmental units. For the same consideration, ‘tone’ is proposed to replace to ‘frequencies’.

5. Page 15, line 11. “see. In fact,”, typo?

6. Page 15, line 5-7 of the countdown. What do you mean by “The syllable consonants in Korean are 7 of ㅂ/p/, ㄷ/t/, ㄱ/k/, ㅁ/m/, ㄴ/n/, ㅇ/ ŋ, ㄹ/l, while Chinese is only 2 of /-n/ and / ŋ/.”? Please check and confirm. I guess you are mentioning the coda types differences between Korean and Chinese?

Reviewer #2: This paper conducts an experiment exploring the difference of the arousal and valence of emotions felt toward Hangul phonemes between Korean and Chinese women. The conclusions are insightful, however, there are some questions remaining unsolved.

Firstly, the articulation of consonants consists of the place of articulation and the method of articulation. This paper only considers the method of articulation, so the author should clarify why they don’t take the effect of the place of consonants into account.

Secondly, this paper simply using nationality as the factor of cultural differences, however, the Chinese participants in this paper are all living in Korea, the length of time living in Korea may affect the result.

Thirdly, Chinese has at least seven dialects including Cantonese, Min, Hakka, etc. The southern dialects are quite different from Mandarin. In this paper, the authors do not consider the different dialectal backgrounds of the participants or at least not clarify this point. For example, most southern Chinese dialects have -m, -n, -ŋ, -p, -t, -k coda. So, in the discussion section, the discussions on the differences of Korean and Chinese is only true for Mandarin Chinese.

Fourthly, since the experiment was conducted online, the outside environments are not the same for all participants. This will somehow affect the result since the researchers can’t clearly see how the participants finish the experiments.

Are there any data of participants are invalid and how the distinguish?

In sum, the experiment done in this paper has shed some lights on the difference of emotions felt toward Hangul phonemes between Korean and Chinese women. The authors should control the experiment environments, and consider more about the dialectal backgrounds of participants, so as to draw a rigorous conclusion.

Typo errors:

p10: “Cultural differences can affect the perception of morphemes with meaning (eg, happiness, anger, fear), but the smallest sound unit that does not contain meaning (eg, /ka/, /nu/, /si/) It is easy to expect that cultural differences will not have an impact on However”

6. PLOS authors have the option to publish the peer review history of their article (what does this mean?). If published, this will include your full peer review and any attached files.

Reviewer #1: No

Reviewer #2: **Yes: **Jingfen, Zhang

---

## [Author Response · Author response to Decision Letter 0]

13 Dec 2022

Editor’s requirement

A) This paper was written by referring to the writing method and form. 

2. Please provide additional details regarding participant consent. In the ethics statement in the Methods and online submission information, please ensure that you have specified (1) whether consent was informed and (2) what type you obtained (for instance, written or verbal, and if verbal, how it was documented and witnessed). If your study included minors, state whether you obtained consent from parents or guardians. If the need for consent was waived by the ethics committee, please include this information. If you are reporting a retrospective study of medical records or archived samples, please ensure that you have discussed whether all data were fully anonymized before you accessed them and/or whether the IRB or ethics committee waived the requirement for informed consent. If patients provided informed written consent to have data from their medical records used in research, please include this information. 

A) We mentioned about the participant consent in procedure part as following: All participants were able to start an online survey after agreeing to participate in the survey. In the first page of online survey form, they were informed about the research procedure, the possible risks, and the right to voluntarily stop participating in the experiment in writing.

3. Thank you for stating the following financial disclosure:“This work was supported by the Ministry of Education of the Republic of Korea and the National Research Foundation of Korea(NRF-2019S1A5C2A04082405)” Please state what role the funders took in the study. If the funders had no role, please state: "The funders had no role in study design, data collection and analysis, decision to publish, or preparation of the manuscript." If this statement is not correct you must amend it as needed. Please include this amended Role of Funder statement in your cover letter; we will change the online submission form on your behalf.

A) We deleted funding statement in manuscript. And we mentioned this information in revised cover letter. 

4. Thank you for stating the following in the Acknowledgments Section of your manuscript: “This work was supported by the Ministry of Education of the Republic of Korea and the National Research Foundation of Korea (NRF-2019S1A5C2A04082405)”. We note that you have provided additional information within the Acknowledgements Section that is not currently declared in your Funding Statement. Please note that funding information should not appear in the Acknowledgments section or other areas of your manuscript. We will only publish funding information present in the Funding Statement section of the online submission form. Please remove any funding-related text from the manuscript and let us know how you would like to update your Funding Statement. Currently, your Funding Statement reads as follows: “This research was supported by the Ministry of Education of the Republic of Korea and the National Research Foundation of Korea (NRF-2019S1A5C2A04082405)”. Please include your amended statements within your cover letter; we will change the online submission form on your behalf. 

A) We deleted funding statement in manuscript. And we mentioned this information in revised cover letter. 

5. We note that you have indicated that data from this study are available upon request. PLOS only allows data to be available upon request if there are legal or ethical restrictions on sharing data publicly. For more information on unacceptable data access restrictions, please see http://journals.plos.org/plosone/s/data-availability#loc-unacceptable-data-access-restrictions. In your revised cover letter, please address the following prompts: a) If there are ethical or legal restrictions on sharing a de-identified data set, please explain them in detail (e.g., data contain potentially sensitive information, data are owned by a third-party organization, etc.) and who has imposed them (e.g., an ethics committee). Please also provide contact information for a data access committee, ethics committee, or other institutional body to which data requests may be sent. b) If there are no restrictions, please upload the minimal anonymized data set necessary to replicate your study findings as either Supporting Information files or to a stable, public repository and provide us with the relevant URLs, DOIs, or accession numbers. For a list of acceptable repositories, please see http://journals.plos.org/plosone/s/data-availability#loc-recommended-repositories. We will update your Data Availability statement on your behalf to reflect the information you provide.

A) There are no restrictions. And we mentioned it in revised cover letter. 

A) The corresponding author already has an ORCID iD. 

7. We note that you have referenced (ie. Bewick et al. [5]) which has currently not yet been accepted for publication. Please remove this from your References and amend this to state in the body of your manuscript: (ie “Bewick et al. [Unpublished]”) as detailed online in our guide for authors http://journals.plos.org/plosone/s/submission-guidelines#loc-reference-style

A) We didn’t have referenced Bewick et al., study. 

Response to the Reviewer #1

1. In the abstract and other part of your paper, I suggest to use consonant, instead of articulation or articulation methods to avoid confusion. In the field of Phonetics Phonology, the term ‘articulation’ refers to the configuration which produces the sounds of speech while both consonant and vowel belong to phoneme category.

A) As you pointed out, we use the word “consonant” instead of articulation or articulation methods. 

2. Page 2, line 4 of the countdown. “contain meaning (eg, /ka/, /nu/, /si/)”, punctuation missing?

A) We corrected the missing punctuation. 

3. 3. Page 2, line 3 of the countdown. “impact on However,”, typo?

A) We corrected the wrong upper case letter(However) to lower case letter(however).

4. Page 4, line 8-9 of the countdown. From a linguistic point of view, it is more appropriate to categorized them with vowel, consonant and tone). Articulation methods seem strange. You are suggested not to use this term as ‘consonant’ is more accurate, in accordance with vowel and other segmental or supra-segmental units. For the same consideration, ‘tone’ is proposed to replace to ‘frequencies’.

A) As you pointed out, we modified “consonant and tone” to replace to “articulation and frequencies”. 

5. Page 15, line 11. “see. In fact,”, typo?

A) We deleted the typo error. 

6. Page 15, line 5-7 of the countdown. What do you mean by “The syllable consonants in Korean are 7 of ㅂ/p/, ㄷ/t/, ㄱ/k/, ㅁ/m/, ㄴ/n/, ㅇ/ ŋ, ㄹ/l, while Chinese is only 2 of /-n/ and / ŋ/.”? Please check and confirm. I guess you are mentioning the coda types differences between Korean and Chinese?

A) We appreciate for pointing out the error. We modified the sentence to convey the exact content. 

Response to the Reviewer #2

1. the articulation of consonants consists of the place of articulation and the method of articulation. This paper only considers the method of articulation, so the author should clarify why they don’t take the effect of the place of consonants into account.

A) Because of the classification of consonants varies depending on the language, we considered only plain consonant, aspirated consonant, and voiced consonant which is a common consonant in many languages and known to give a similar emotional impression in this study. The concepts that were not fully considered in this study were mentioned as limitations of the study. 

2. this paper simply using nationality as the factor of cultural differences, however, the Chinese participants in this paper are all living in Korea, the length of time living in Korea may affect the result.

A) This study was conducted on Chinese people who came to live in Korea after entering a Korean university. They are Chinese who have been introduced to Korean culture since they became adults, it is assumed that they have a unique cultural characteristic acquired in the process of growing up even if they have become familiar with Korean culture. However, as you pointed out, all the levels of Korean cultural experience of each participant had not controlled. But this study is the first study to compare the difference in perception between arousal and valence values for Korean language, so we discussed it in terms of limitations so that it can be fully considered in future studies. 

3. Chinese has at least seven dialects including Cantonese, Min, Hakka, etc. The southern dialects are quite different from Mandarin. In this paper, the authors do not consider the different dialectal backgrounds of the participants or at least not clarify this point. For example, most southern Chinese dialects have -m, -n, -ŋ, -p, -t, -k coda. So, in the discussion section, the discussions on the differences of Korean and Chinese is only true for Mandarin Chinese.

A) In this study, only Chinese who use Mandarin were required to response to the questionnaire when distributing the questionnaire. This was not specified in the paper, so the relevant content was written reflecting what you pointed out. 

4. since the experiment was conducted online, the outside environments are not the same for all participants. This will somehow affect the result since the researchers can’t clearly see how the participants finish the experiments. Are there any data of participants are invalid and how the distinguish? The experiment done in this paper has shed some lights on the difference of emotions felt toward Hangul phonemes between Korean and Chinese women. The authors should control the experiment environments, and consider more about the dialectal backgrounds of participants, so as to draw a rigorous conclusion. 

A) Although the experiment was conducted online, the environment of all participants was not controlled equally, but we emphasized the notice of the main points before participating in the experiment as follows: 1) Set the time to run the experiment for 1 hour (do not interrupt this experiment in the middle), 2) the surrounding environment is set so that it is not affected by external stimuli (e.g., noise), 3) recommend using earphones and check the computer volume of device in advance. 

Since this study was intended to measure the emotional response to auditory stimuli presented, it was assumed that the participants’ basic emotional state would have a more significant influence, although the surrounding environmental factors were important at the time of the experiment. Accordingly, the emotional state of each participant was measured in advance, and as a result, it was confirmed that there was no difference in the emotional state of the participants. 

5. Typo errors: p10: “Cultural differences can affect the perception of morphemes with meaning (eg, happiness, anger, fear), but the smallest sound unit that does not contain meaning (eg, /ka/, /nu/, /si/) It is easy to expect that cultural differences will not have an impact on However”

A) We corrected as “Cultural differences can affect the perception of morphemes with meaning (eg, happiness, anger, fear), but the smallest sound unit that does not contain meaning (eg, /ka/, /nu/, /si/). It is easy to expect that cultural differences will not have an impact on however,”

---

## [Decision Letter · Decision Letter 1]

16 Jan 2023

PONE-D-22-15241R1Differences in Arousal and Valence on the Korean Phoneme of Artificial Voice between Korean and Chinese WomenPLOS ONE

Dear Jang-Han Lee,

Thank you for submitting your manuscript to PLOS ONE. After careful consideration, we feel that it has merit but does not fully meet PLOS ONE’s publication criteria as it currently stands. Therefore, we invite you to submit a revised version of the manuscript that addresses the points raised during the review process.

We look forward to receiving your revised manuscript.

Kind regards,

Tai Ming Wut

Academic Editor

PLOS ONE

Journal Requirements:

Reviewers' comments:

Reviewer's Responses to Questions

**Comments to the Author**

1. If the authors have adequately addressed your comments raised in a previous round of review and you feel that this manuscript is now acceptable for publication, you may indicate that here to bypass the “Comments to the Author” section, enter your conflict of interest statement in the “Confidential to Editor” section, and submit your "Accept" recommendation.

Reviewer #2: (No Response)

2. Is the manuscript technically sound, and do the data support the conclusions?

Reviewer #2: Partly

3. Has the statistical analysis been performed appropriately and rigorously? 

Reviewer #2: Yes

4. Have the authors made all data underlying the findings in their manuscript fully available?

Reviewer #2: Yes

5. Is the manuscript presented in an intelligible fashion and written in standard English?

Reviewer #2: Yes

6. Review Comments to the Author

Reviewer #2: 1. There is no voiced consonant in Mandarin Chinese, the result find in the paper shows that ‘line 216-218 Korean reported higher arousal than Chinese in lenis and aspirated consonant, whereas there was no distinct difference between nationalities in voiced consonant’. So, the phonetic system (inventories) of Mandarin may affect the reaction of Chinese women in this experiment, since lenis and voiced consonants are only phonetic variants, but not phoneme in Mandarin. I think this point is more pivotal than the differences between end sounds in Korean and Chinese as the author mention. But in the discussion part, the authors do not mention the lack of voiced consonant in Mandarin.

2. Although the authors mention three limitations at last, we should be aware that all these limitations make the present paper less convincing. I suggest the authors to conduct the same experiment on male participants and see whether the result is the same with the women’.

Below is one confusing sentence and the authors need to double check.

(1) p3 line 58, For example, it is assumed that the word for dog in thousands of languages will use consonants and vowels to symbolize large entities, and consonants and vowels to symbolize relatively small entities for ants.

Some typos:

(1) p4 line 80, According to the existing claim that all sounds have acoustic characteristics such as ‘rough or soft’, strong or weak’… [only one ’ ]

(2) p8 line 157, measures ''trait anxiety'', it is better to use ‘’ in English instead of “”.

7. PLOS authors have the option to publish the peer review history of their article (what does this mean?). If published, this will include your full peer review and any attached files.

Reviewer #2: No

---

## [Author Response · Author response to Decision Letter 1]

28 Feb 2023

We appreciate your sincere comments. 

1. There is no voiced consonant in Mandarin Chinese, the result find in the paper shows that ‘line 216-218 Korean reported higher arousal than Chinese in lenis and aspirated consonant, whereas there was no distinct difference between nationalities in voiced consonant’. So, the phonetic system (inventories) of Mandarin may affect the reaction of Chinese women in this experiment, since lenis and voiced consonants are only phonetic variants, but not phoneme in Mandarin. I think this point is more pivotal than the differences between end sounds in Korean and Chinese as the author mention. But in the discussion part, the authors do not mention the lack of voiced consonant in Mandarin.

A) We added discussion as followed: Koreans showed higher arousal than Chinese in lenis and aspirated consonants, while there was no clear difference between nationalities in voiced consonants. In general, Mandarin Chinese does not use voiced and voiceless sounds as discriminator as in Korean, so how well Chinese and Korean distinguish between voiced and aspirated sounds does not seem to affect the valence of audible sounds. However, the response values to the voiced consonants collected in this study tended to be somewhat lower in both Koreans and Chinese compared to lenis and aspirated consonants. In other words, it is estimated that the sound accompanying the vocal cords like voiced consonants lowered the arousal level, while it played a role in activating valence level. 

2. Although the authors mention three limitations at last, we should be aware that all these limitations make the present paper less convincing. I suggest the authors to conduct the same experiment on male participants and see whether the result is the same with the women’.

A) We appreciate for your sincere opinion. In this study, only one gender was selected that could be considered similar in response level or pattern, assuming that there may be a difference in response according to experience depending on the emotional state. Therefore, we tried to match the homogeneity of the emotional state or level within the group as much as possible. However, as we also pointed out in our paper, we will conduct further studies on men with the same experiment, since it is assumed that gender differences may exist according to cultures. We are currently preparing for a specific experiment and will implement it soon. 

3. Below is one confusing sentence and the authors need to double check. (1) p3 line 58, For example, it is assumed that the word for dog in thousands of languages will use consonants and vowels to symbolize large entities, and consonants and vowels to symbolize relatively small entities for ants.

A) We modified the sentence as: For example, in thousands of languages, the word ‘dog’ tends to use consonants and vowels with sounds that can represent relatively large entities, while consonants and vowels with smaller sounds can be used to symbolize relatively small entities, such as ‘ant’. 

4. Some typos: (1) p4 line 80, According to the existing claim that all sounds have acoustic characteristics such as ‘rough or soft’, strong or weak’… [only one ’ ] (2) p8 line 157, measures ''trait anxiety'', it is better to use ‘’ in English instead of “”.

A) We corrected all the typos you found.

---

## [Decision Letter · Decision Letter 2]

22 Mar 2023

Differences in Arousal and Valence on the Korean Phoneme of Artificial Voice between Korean and Chinese Women

PONE-D-22-15241R2

Dear Jang-Han Lee,

We’re pleased to inform you that your manuscript has been judged scientifically suitable for publication and will be formally accepted for publication once it meets all outstanding technical requirements.

Kind regards,

Tai Ming Wut

Academic Editor

PLOS ONE

Additional Editor Comments (optional):

Reviewers' comments:

Reviewer's Responses to Questions

**Comments to the Author**

1. If the authors have adequately addressed your comments raised in a previous round of review and you feel that this manuscript is now acceptable for publication, you may indicate that here to bypass the “Comments to the Author” section, enter your conflict of interest statement in the “Confidential to Editor” section, and submit your "Accept" recommendation.

Reviewer #1: All comments have been addressed

Reviewer #2: All comments have been addressed

2. Is the manuscript technically sound, and do the data support the conclusions?

Reviewer #1: Yes

Reviewer #2: Yes

3. Has the statistical analysis been performed appropriately and rigorously? 

Reviewer #1: I Don't Know

Reviewer #2: Yes

4. Have the authors made all data underlying the findings in their manuscript fully available?

Reviewer #1: Yes

Reviewer #2: Yes

5. Is the manuscript presented in an intelligible fashion and written in standard English?

Reviewer #1: Yes

Reviewer #2: Yes

6. Review Comments to the Author

Reviewer #1: In this study, the difference between the arousal and valence of emotions felt toward Hangul phonemes was explored according to consonant and vowel through comparison between Korean and Chinese women. This is an interesting study and the authors have collected a unique dataset using rigorous methodology. The design of two groups of participants from different language backgrounds makes the experiment reasonable for the purpose. I believe the authors conducted the experiment carefully in line with the strict procedure. Overall the paper is generally well written and structured. Thus, the current study is a topic of relevance and general interest to the reader of the journal.

Reviewer #2: The authors have already answered my questions and they have corrected all the typos. So I have no any further questions.

7. PLOS authors have the option to publish the peer review history of their article (what does this mean?). If published, this will include your full peer review and any attached files.

Reviewer #1: No

Reviewer #2: No

---

## [Editor Report · Acceptance letter]

29 Mar 2023

PONE-D-22-15241R2 

Differences in Arousal and Valence on the Korean Phoneme of Artificial Voice between Korean and Chinese Women 

Dear Dr. Lee:

I'm pleased to inform you that your manuscript has been deemed suitable for publication in PLOS ONE. Congratulations! Your manuscript is now with our production department. 

Kind regards, 

on behalf of

Dr. Tai Ming Wut 

Academic Editor

PLOS ONE